# TRAJECTORY PREDICTION USING EQUIVARIANT CONTINUOUS CONVOLUTION

**Robin Walters** [*]
Northeastern University
r.walters@northeastern.edu

**Jinxi Li**[*]
Northeastern University
li.jinxi1@northeastern.edu

**Rose Yu**
University of California, San Diego
roseyu@ucsd.edu

## ABSTRACT

Trajectory prediction is a critical part of many AI applications, for example, the safe operation of autonomous vehicles. However, current methods are prone to making inconsistent and physically unrealistic predictions. We leverage insights from fluid dynamics to overcome this limitation by considering internal symmetry in real-world trajectories. We propose a novel model, **E**quivariant **C**ontinous **CO**nvolution (ECCO) for improved trajectory prediction. ECCO uses rotationally-equivariant continuous convolutions to embed the symmetries of the system. On both vehicle and pedestrian trajectory datasets, ECCO attains competitive accuracy with significantly fewer parameters. It is also more sample efficient, generalizing automatically from few data points in any orientation. Lastly, ECCO improves generalization with equivariance, resulting in more physically consistent predictions. Our method provides a fresh perspective towards increasing trust and transparency in deep learning models. Our code and data can be found at https://github.com/Rose-STL-Lab/ECCO.

## 1 INTRODUCTION

Trajectory prediction is one of the core tasks in AI, from the movement of basketball players to fluid particles to car traffic (Sanchez-Gonzalez et al., 2020; Gao et al., 2020; Shah & Romijnders, 2016). A common abstraction underlying these tasks is the movement of many interacting agents, analogous to a many-particle system. Therefore, understanding the states of these particles, their dynamics, and hidden interactions is critical to accurate and robust trajectory forecasting.

Even for purely physical systems such as in particle physics, the complex interactions among a large number of particles makes this a difficult problem. For vehicle or pedestrian trajectories, this challenge is further compounded with latent factors such as human psychology. Given these difficulties, current approaches require large amounts of training data and many model parameters. State-of-the-art methods in this domain such as Gao et al. (2020) are based on graph neural networks. They do not exploit the physical properties of system and often make predictions which are not self-consistent or physically meaningful. Furthermore, they predict a single agent trajectory at a time instead of multiple agents simultaneously.

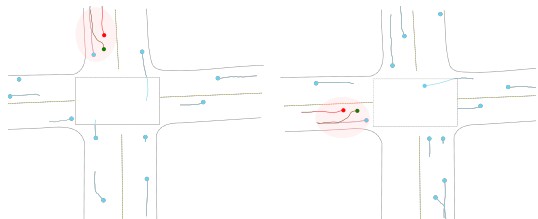

Figure 1: Car trajectories in two scenes. Though the entire scenes are not related by a rotation, the circled areas are. ECCO exploits this symmetry to improve generalization and sample efficiency.

---

[*]Equal Contribution

Our model is built upon a key insight of many-particle systems pertaining to intricate internal symmetry. Consider a model which predicts the trajectory of cars on a road. To be successful, such a model must understand the physical behavior of vehicles together with human psychology. It should distinguish left from right turns, and give consistent outputs for intersections rotated with different orientation. As shown in Figure 1, a driver's velocity rotates with the entire scene, whereas vehicle interactions are invariant to such a rotation. Likewise, psychological factors such as reaction speed or attention may be considered vectors with prescribed transformation properties. Data augmentation is a common practice to deal with rotational invariance, but it cannot guarantee invariance and requires longer training. Since rotation is a continuous group, augmentation requires sampling from infinitely many possible angles.

In this paper, we propose an equivariant continuous convolutional model, ECCO, for trajectory forecasting. Continuous convolution generalizes discrete convolution and is adapted to data in many-particle systems with complex local interactions. Ummenhofer et al. (2019) designed a model using continuous convolutions for particle-based fluid simulations. Meanwhile, equivariance to group symmetries has proven to be a powerful tool to integrate physical intuition in physical science applications (Wang et al., 2020; Brown & Lunter, 2019; Kanwar et al., 2020). Here, we test the hypothesis that an equivariant model can also capture internal symmetry in non-physical human behavior. Our model utilizes a novel weight sharing scheme, torus kernels, and is rotationally equivariant.

We evaluate our model on two real-world trajectory datasets: Argoverse autonomous vehicle dataset (Chang et al., 2019) and TrajNet++ pedestrian trajectory forecasting challenge (Kothari et al., 2020). We demonstrate on par or better prediction accuracy to baseline models and data augmentation with fewer parameters, better sample efficiency, and stronger generalization properties. Lastly, we demonstrate theoretically and experimentally that our polar coordinate-indexed filters have lower equivariance discretization error due to being better adapted to the symmetry group.

Our main contributions are as follows:

- We propose **E**quivariant **C**ontinous **CO**nvolution (ECCO), a rotationally equivariant deep neural network that can capture internal symmetry in trajectories.

- We design ECCO using a novel weight sharing scheme based on orbit decomposition and polar coordinate-indexed filters. We implement equivariance for both the standard and regular representation $L^2(\mathrm{SO}(2))$.

- On benchmark Argoverse and TrajNet++ datasets, ECCO demonstrates comparable accuracy while enjoying better generalization, fewer parameters, and better sample complexity.

## 2 RELATED WORK

**Trajectory Forecasting**    For vehicle trajectories, classic models in transportation include the Car-Following model (Pipes, 1966) and Intelligent Driver model (Kesting et al., 2010). Deep learning has also received considerable attention; for example, Liang et al. (2020) and Gao et al. (2020) use graph neural networks to predict vehicle trajectories. Djuric et al. (2018) use rasterizations of the scene with CNN. See the review paper by Veres & Moussa (2019) for deep learning in transportation. For human trajectory modeling, Alahi et al. (2016) propose Social LSTM to learn these human-human interactions. TrajNet (Sadeghian et al., 2018) and TrajNet++ (Kothari et al., 2020) introduce benchmarking for human trajectory forecasting. We refer readers to Rudenko et al. (2020) for a comprehensive survey. Nevertheless, many deep learning models are data-driven. They require large amounts of data, have many parameters, and can generate physically inconsistent predictions.

**Continuous Convolution**    Continuous convolutions over point clouds (CtsConv) have been successfully applied to classification and segmentation tasks (Wang et al., 2018; Lei et al., 2019; Xu et al., 2018; Wu et al., 2019; Su et al., 2018; Li et al., 2018; Hermosilla et al., 2018; Atzmon et al., 2018; Hua et al., 2018). More recently, a few works have used continuous convolution for modeling trajectories or flows. For instance, Wang et al. (2018) uses CtsConv for inferring flow on LIDAR data. Schenck & Fox (2018) and Ummenhofer et al. (2019) model fluid simulation using CtsConv. Closely related to our work is Ummenhofer et al. (2019), who design a continuous convolution network for particle-based fluid simulations. However, they use a ball-to-sphere mapping which is not well-adapted for rotational equivariance and only encode 3 frames of input. Graph neural networks (GNNs) are a related strategy which have been used for modeling particle system

dynamics (Sanchez-Gonzalez et al., 2020). GNNs are also permutation invariant, but they do not natively encode relative positions and local interaction as a CtsConv-based network does.

**Equivariant and Invariant Deep Learning** Developing neural nets that preserve symmetries has been a fundamental task in image recognition (Cohen et al., 2019b; Weiler & Cesa, 2019; Cohen & Welling, 2016a; Chidester et al., 2018; Lenc & Vedaldi, 2015; Kondor & Trivedi, 2018; Bao & Song, 2019; Worrall et al., 2017; Cohen & Welling, 2016b; Weiler et al., 2018; Dieleman et al., 2016; Maron et al., 2020). Equivariant networks have also been used to predict dynamics: for example, Wang et al. (2020) predicts fluid flow using Galilean equivariance but only for gridded data. Fuchs et al. (2020) use $SE(3)$-equivariant transformers to predict trajectories for a small number of particles as a regression task. As in this paper, both Bekkers (2020) and Finzi et al. (2020) address the challenge of parameterizing a kernel over continuous Lie groups. Finzi et al. (2020) apply their method to trajectory prediction on point clouds using a small number of points following strict physical laws. Worrall et al. (2017) also parameterizes convolutional kernels using polar coordinates, but maps these onto a rectilinear grid for application to image data. Weng et al. (2018) address rotational equivariance by inferring a global canonicalization of the input. Similar to our work, Esteves et al. (2018) use functions evenly sampled on the circle, however, their features are only at a single point whereas we assign feature vectors to each point in a point cloud. Thomas et al. (2018) introduce Tensor Field Networks which are $SO(3)$-equivariant continuous convolutions. Unlike our work, both Worrall et al. (2017) and Thomas et al. (2018) define their kernels using harmonic functions. Our weight sharing method using orbits and stabilizers is simpler as it does not require harmonic functions or Clebsch-Gordon coefficients. Unlike previous work, we implement a regular representation for the continuous rotation group $SO(2)$ which is compatible with pointwise non-linearities and enjoys an empirical advantage over irreducible representations.

## 3 BACKGROUND

We first review the necessary background of continuous convolution and rotational equivariance.

### 3.1 CONTINUOUS CONVOLUTION

Continuous convolution (`CtsConv`) generalizes the discrete convolution to point clouds. It provides an efficient and spatially aware way to model the interactions of nearby particles. Let $\mathbf{f}^{(i)} \in \mathbb{R}^{c_{\text{in}}}$ denote the feature vector of particle $i$. Thus $\mathbf{f}$ is a vector field which assigns to the points $\mathbf{x}^{(i)}$ a vector in $\mathbb{R}^{c_{\text{in}}}$. The kernel of the convolution $K \colon \mathbb{R}^2 \to \mathbb{R}^{c_{\text{out}} \times c_{\text{in}}}$ is a *matrix* field: for each point $\mathbf{x} \in \mathbb{R}^2$, $K(\mathbf{x})$ is a $c_{\text{out}} \times c_{\text{in}}$ matrix. Let $a$ be a radial local attention map with $a(r) = 0$ for $r > R$. The output feature vector $\mathbf{g}^{(i)}$ of particle $i$ from the continous convolution is given by

$$\mathbf{g}^{(i)} = \text{CtsConv}_{K,R}(\mathbf{x}, \mathbf{f}; \mathbf{x}^{(i)}) = \sum_j a(\|\mathbf{x}^{(j)} - \mathbf{x}^{(i)}\|) K(\mathbf{x}^{(j)} - \mathbf{x}^{(i)}) \cdot \mathbf{f}^{(j)}. \tag{1}$$

`CtsConv` is naturally equivariant to permutation of labels and is translation invariant. Equation 1 is closely related to graph neural network (GNN) (Kipf & Welling, 2017; Battaglia et al., 2018), which is also permutation invariant. Here the graph is dynamic and implicit with nodes $\mathbf{x}^{(i)}$ and edges $e_{ij}$ if $\|\mathbf{x}^{(i)} - \mathbf{x}^{(j)}\| < R$. Unlike a GNN which applies the same weights to all neighbours, the kernel $K$ depends on the relative position vector $\mathbf{x}^{(i)} - \mathbf{x}^{(j)}$.

### 3.2 ROTATIONAL EQUIVARIANCE

Continuous convolution is not naturally rotationally equivariant. Fortunately, we can translate the technique of rotational equivariance on CNNs to continuous convolutions. We use the language of Lie groups and their representations. For more background, see Hall (2015) and Knapp (2013).

More precisely, we denote the symmetry group of 2D rotations by $SO(2) = \{\text{Rot}_\theta : 0 \le \theta < 2\pi\}$. As a Lie group, it has both a group structure $\text{Rot}_{\theta_1} \circ \text{Rot}_{\theta_2} = \text{Rot}_{(\theta_1 + \theta_2) \bmod 2\pi}$ which a continuous map with respect to the topological structure. As a manifold, $SO(2)$ is homomeomorphic to the circle $S^1 \cong \{\mathbf{x} \in \mathbb{R}^2 : \|\mathbf{x}\| = 1\}$. The group $SO(2)$ can act on a vector space $\mathbb{R}^c$ by specifying a *representation* map $\rho \colon SO(2) \to GL(\mathbb{R}^c)$ which assigns to each element of $SO(2)$ an element of the set of invertible $c \times c$ matrices $GL(\mathbb{R}^c)$. The map $\rho$ must a be homomorphism

$\rho(\text{Rot}_{\theta_1})\rho(\text{Rot}_{\theta_1}) = \rho(\text{Rot}_{\theta_1} \circ \text{Rot}_{\theta_2})$. For example, the *standard representation* $\rho_1$ on $\mathbb{R}^2$ is by $2 \times 2$ rotation matrices. The *regular representation* $\rho_{\text{reg}}$ on $L^2(\text{SO}(2)) = \{\varphi : \text{SO}(2) \rightarrow \mathbb{R} : |\varphi|^2 \text{ is integrable}\}$ is $\rho_{\text{reg}}(\text{Rot}_\phi)(\varphi) = \varphi \circ \text{Rot}_{-\phi}$. Given input $\mathbf{f}$ with representation $\rho_{\text{in}}$ and output with representation $\rho_{\text{out}}$, a map $F$ is SO(2)-equivariant if

$$F(\rho_{\text{in}}(\text{Rot}_\theta)\mathbf{f}) = \rho_{\text{out}}(\text{Rot}_\theta)F(\mathbf{f}).$$

Discrete CNNs are equivariant to a group $G$ if the input, output, and hidden layers carry a $G$-action and the linear layers and activation functions are all equivariant (Kondor & Trivedi, 2018). One method for constructing equivariant discrete CNNs is steerable CNN (Cohen & Welling, 2016b). Cohen et al. (2019a) derive a general constraint for when a convolutional kernel $K : \mathbb{R}^b \rightarrow \mathbb{R}^{c_{\text{out}} \times c_{\text{in}}}$ is $G$-equivariant. Assume $G$ acts on $\mathbb{R}^b$ and that $\mathbb{R}^{c_{\text{out}}}$ and $\mathbb{R}^{c_{\text{in}}}$ are $G$-representations $\rho_{\text{out}}$ and $\rho_{\text{in}}$ respectively, then $K$ is $G$-equivariant if for all $g \in G, \mathbf{x} \in \mathbb{R}^2$,

$$K(g\mathbf{x}) = \rho_{\text{out}}(g)K(\mathbf{x})\rho_{\text{in}}(g^{-1}). \tag{2}$$

For the group SO(2), Weiler & Cesa (2019) solve this constraint using circular harmonic functions to give a basis of discrete equivariant kernels. In contrast, our method is much simpler and uses orbits and stabilizers to create continuous convolution kernels.

## 4    ECCO: TRAJECTORY PREDICTION USING ROTATIONALLY EQUIVARIANT CONTINUOUS CONVOLUTION

In trajectory prediction, given historical position and velocity data of $n$ particles over $t_{\text{in}}$ timesteps, we want to predict their positions over the next $t_{\text{out}}$ timesteps. Denote the ground truth dynamics as $\xi$, which maps $\xi(\mathbf{x}_{t-t_{\text{in}}:t}, \mathbf{v}_{t-t_{\text{in}}:t}) = \mathbf{x}_{t:t+t_{\text{out}}}$. Motivated by the observation in Figure 1, we wish to learn a model $f$ that approximates the underlying dynamics while preserving the internal symmetry in the data, specifically rotational equivariance.

We introduce ECCO, a model for trajectory prediction based on rotationally equivariant continuous convolution. We implement rotationally equivariant continuous convolutions using a weight sharing scheme based on orbit decomposition. We also describe equivariant per-particle linear layers which are a special case of continuous convolution with radius $R = 0$ analogous to 1x1 convolutions in CNNs. Such layers are useful for passing information between layers from each particle to itself.

### 4.1    ECCO MODEL OVERVIEW

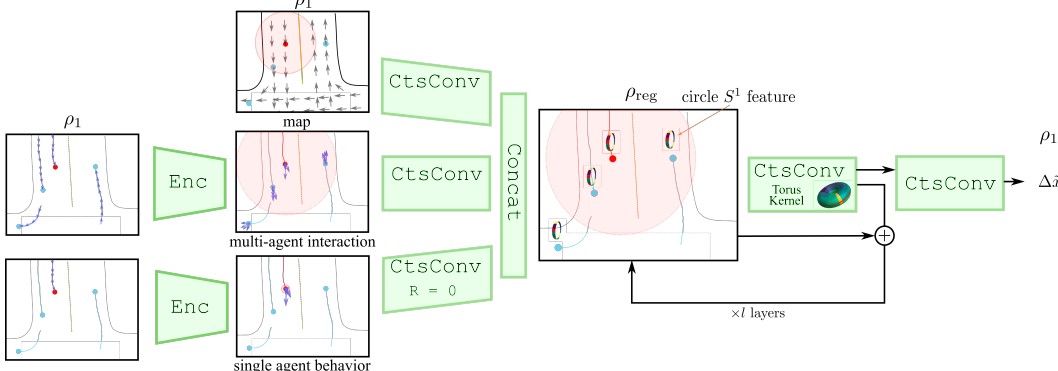

Figure 2: Overview of model architecture. Past velocities are aggregated by an encoder `Enc`. Together with map information this is then encoded by 3 `CtsConv`s into $\rho_{\text{reg}}$ features. Then $l + 1$ `CtsConv` layers are used to predict $\Delta \tilde{x}$. The predicted position $\hat{x}_{t+1} = \Delta \tilde{x} + \tilde{x}$ where $\tilde{x}$ is a numerically extrapolated using velocity and accleration. Since $\Delta \tilde{x}$ is translation invariant, $\hat{x}$ is equivariant.

The high-level architecture of ECCO is illustrated in Figure 2. It is important to remember that the input, output, and hidden layers are all vector fields over the particles. Oftentimes, there is also

environmental information available in the form of road lane markers. Denote marker positions by $\mathbf{x}_{\text{map}}$ and direction vectors by $\mathbf{v}_{\text{map}}$. This data is thus also a particle field, but static.

To design an equivariant network, one must choose the group representation. This choice plays an important role in shaping the learned hidden states. We focus on two representations of SO(2): $\rho_1$ and $\rho_{\text{reg}}$. The representation $\rho_1$ is that of our input features, and $\rho_{\text{reg}}$ is for the hidden layers. For $\rho_1$, we constrain the kernel in Equation 1. For $\rho_{\text{reg}}$, we further introduce a new operator, convolution with torus kernels.

In order to make continuous convolution rotationally equivariant, we translate the general condition for discrete CNNs developed in Weiler & Cesa (2019) to continuous convolution. We define the convolution kernel $K$ in polar coordinates $K(\theta, r)$. Let $\mathbb{R}^{c_{\text{out}}}$ and $\mathbb{R}^{c_{\text{in}}}$ be SO(2)-representations $\rho_{\text{out}}$ and $\rho_{\text{in}}$ respectively, then the equivariance condition requires the kernel to satisfy

$$K(\theta + \phi, r) = \rho_{\text{out}}(\text{Rot}_\theta)K(\phi, r)\rho_{\text{in}}(\text{Rot}_\theta^{-1}). \qquad (3)$$

Imposing such a constraint for continuous convolution requires us to develop an efficient weight sharing scheme for the kernels, which solve Equation 3.

## 4.2 WEIGHT SHARING BY ORBITS AND STABILIZERS.

Given a point $\mathbf{x} \in \mathbb{R}^2$ and a group $G$, the set $O_{\mathbf{x}} = \{g\mathbf{x} : g \in G\}$ is the *orbit* of the point $\mathbf{x}$. The set of orbits gives a partition of $\mathbb{R}^2$ into the origin and circles of radius $r > 0$. The set of group elements $G_{\mathbf{x}} = \{g : g\mathbf{x} = x\}$ fixing $\mathbf{x}$ is called the *stabilizer* of the point $\mathbf{x}$. We use the orbits and stabilizers to constrain the weights of $K$. Simply put, we share weights across orbits and constrain weights according to stabilizers, as shown in Figure 3-Left.

The ray $D = \{(0, r) : r \geq 0\}$ is a *fundamental domain* for the action of $G = \text{SO}(2)$ on base space $\mathbb{R}^2$. That is, $D$ contains exactly one point from each orbit. We first define $K(0, r)$ for each $(0, r) \in D$. Then we compute $K(\theta, r)$ from $K(0, r)$ by setting $\phi = 0$ in Equation 3 as such

$$K(\theta, r) = \rho_{\text{out}}(\text{Rot}_\theta)K(0, r)\rho_{\text{in}}(\text{Rot}_\theta^{-1}). \qquad (4)$$

For $r > 0$, the group acts *freely* on $(0, r)$, i.e. the stabilizer contains only the identity. This means that Equation 3 imposes no additional constraints on $K(0, r)$. Thus $K(0, r) \in \mathbb{R}^{c_{\text{out}} \times c_{\text{in}}}$ is a matrix of freely learnable weights.

For $r = 0$, however, the orbit $O_{(0,0)}$ is only one point. The stabilizer of $(0, 0)$ is all of $G$, which requires

$$K(0, 0) = \rho_{\text{out}}(\text{Rot}_\theta)K(0, 0)\rho_{\text{in}}(\text{Rot}_\theta^{-1}) \text{ for all } \theta. \qquad (5)$$

Thus $K(0, 0)$ is an equivariant per-particle linear map $\rho_{\text{in}} \to \rho_{\text{out}}$.

We can analytically solve Equation 5 for $K(0, 0)$ using representation theory. Table 1 shows the unique solutions for different combinations of $\rho_1$ and $\rho_{\text{reg}}$. For details see subsection A.3.

Table 1: Equivariant linear maps for $K(0, 0)$. Trainable weights are $c \in \mathbb{R}$ and $\kappa \colon S^1 \to \mathbb{R}$, where $S^1$ is the manifold underlying SO(2).

| $\rho_{\text{in}}$ | $\rho_{\text{out}} = \rho_1$ | $\rho_{\text{out}} = \rho_{\text{reg}}$ |
|---|---|---|
| $\rho_1$ | $(a, b) \mapsto (ca, cb)$ | $ca\cos(\theta) + cb\sin(\theta)$ |
| $\rho_{\text{reg}}$ | $\mathbf{f} \mapsto c \begin{pmatrix} \int_{S^1} \mathbf{f}(\theta)\cos(\theta)d\theta \\ \int_{S^1} \mathbf{f}(\theta)\sin(\theta)d\theta \end{pmatrix}$ | $\int_{S^1} \kappa(\theta - \phi)\mathbf{f}(\phi)d\phi$ |

Note that 2D and 3D rotation equivariant continuous convolutions are implemented in Worrall et al. (2017) and Thomas et al. (2018) respectively. They both use harmonic functions which require expensive evaluation of analytic functions at each point. Instead, we provide a simpler solution. We require only knowledge of the orbits, stabilizers, and input/output representations. Additionally, we bypass Clebsch-Gordon decomposition used in Thomas et al. (2018) by mapping directly between the representations in our network. Next, we describe an efficient implementation of equivariant continuous convolution.

## 4.3 POLAR COORDINATE KERNELS

Rotational equivariance informs our kernel discretization and implementation. We store the kernel $K$ of continuous convolution as a 4-dimensional tensor by discretizing the domain. Specifically, we

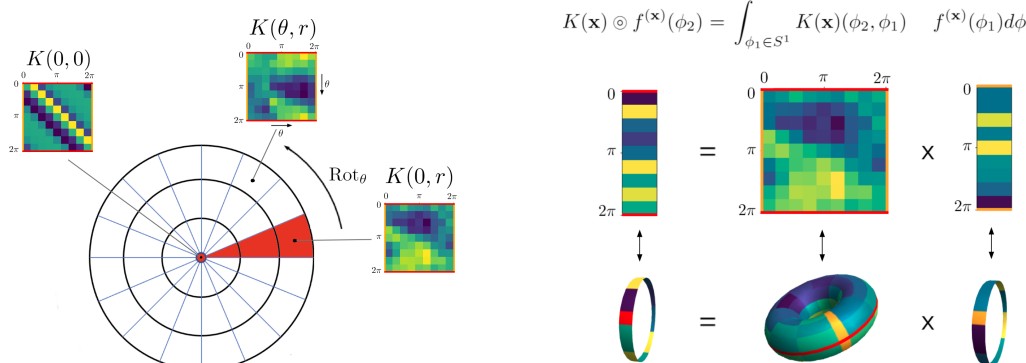

Figure 3: Left: A torus kernel field $K$ from a $\rho_{\text{reg}}$-field to a $\rho_{\text{reg}}$-field. The kernel is itself a field: at each point $\mathbf{x}$ in space the kernel $K(\mathbf{x})$ yields a different matrix. We denote the $(\phi_2, \phi_1)$ entry of the matrix at $\mathbf{x} = (\theta, r)$ by $K(\theta, r)(\phi_2, \phi_1)$. The matrices along the red sector are freely trainable. The matrices at all white sectors are determined by those in the red sector according to the circular shifting rule illustrated above. The matrix at the red bullseye is trainable but constrained to be circulant, i.e. preserved by the circular shifting rule. Right: The torus kernel acts on features which are functions on the circle. By cutting open the torus and features along the reg and orange lines we can identify the operation at each point with matrix multiplication.

discretize $\mathbb{R}^2$ using polar coordinates with $k_\theta$ angular slices and $k_r$ radial steps. We then evaluate $K$ at any $(\theta, r)$ using bilinear interpolation from four closest polar grid points. This method accelerates computation since we do not need to use Equation 4 to repeatedly compute $K(\theta, r)$ from $K(0, r)$. The special case of $K(0, 0)$ results in a polar grid with a "bullseye" at the center (see Figure 3-Left).

We discretize angles finely and radii more coarsely. This choice is inspired by real-world observation that drivers tend to be more sensitive to the angle of an incoming car than its exact distance, Our equivariant kernels are computationally efficient and have very few parameters. Moreover, we will discuss later in Section 4.5 that despite discretization, the use of polar coordinates allows for very low equivariance error.

### 4.4    HIDDEN LAYERS AS REGULAR REPRESENTATIONS

Regular representation $\rho_{\text{reg}}$ has shown better performance than $\rho_1$ for finite groups (Cohen et al., 2019a; Weiler & Cesa, 2019). But the naive $\rho_{\text{reg}} = \{\varphi\colon G \to \mathbb{R}\}$ for an infinite group $G$ is too large to work with. We choose the space of square-integrable functions $L^2(G)$. It contains all irreducible representations of $G$ and is compatible with pointwise non-linearities.

**Discretization.**    However, $L^2(\mathrm{SO}(2))$ is still infinite-dimensional. We resolve this by discretizing the manifold $S^1$ underlying $\mathrm{SO}(2)$ into $k_{\text{reg}}$ even intervals. We represent functions $f \in L^2(\mathrm{SO}(2))$ by the vector of values $[f(\mathrm{Rot}_{2\pi i/k_{\text{reg}}})]_{0 \le i < k_{\text{reg}}}$. We then evaluate $f(\mathrm{Rot}_\theta)$ using interpolation.

We separate the number of angular slices $k_\theta$ and the size of the kernel $k_{\text{reg}}$. If we tie them together and set $k_\theta = k_{\text{reg}}$, this is equivalent to implementing cyclic group $C_{k_{\text{reg}}}$ symmetry with the regular representation. Then increasing $k_\theta$ would also increases $k_{\text{reg}}$, which incurs more parameters.

**Convolution with Torus Kernel.**    In addition to constraining the kernel $K$ of Equation 1 as in $\rho_1$, $\rho_{\text{reg}}$ poses an additional challenge as it is a function on a circle. We introduce a new operator from functions on the circle to functions on the circle called a torus kernel.

First, we replace input feature vectors in $\mathbf{f} \in \mathbb{R}^c$ with elements of $L^2(\mathrm{SO}(2))$. The input feature $\mathbf{f}$ becomes a $\rho_{\text{reg}}$-field, that is, for each $\mathbf{x} \in \mathbb{R}^2$, $\mathbf{f}^{(\mathbf{x})}$ is a real-value function on the circle $S^1 \to \mathbb{R}$. For the kernel $K$, we replace the matrix field with a map $K\colon \mathbb{R}^2 \to \rho_{\text{reg}} \otimes \rho_{\text{reg}}$. Instead of a matrix, $K(\mathbf{x})$ is a map $S^1 \times S^1 \to \mathbb{R}$. Here $(\phi_1, \phi_2) \in S^1 \times S^1$ plays the role of continuous matrix indices and we may consider $K(\mathbf{x})(\phi_1, \phi_2) \in \mathbb{R}$ analogous to a matrix entry. Topologically, $S^1 \times S^1$ is a torus and hence we call $K(\mathbf{x})$ a *torus kernel*. The matrix multiplication $K(\mathbf{x}) \cdot \mathbf{f}^{(\mathbf{x})}$ in Equation 1

must be replaced by the integral transform

$$K(\mathbf{x}) \odot f^{(\mathbf{x})}(\phi_2) = \int_{\phi_1 \in S^1} K(\mathbf{x})(\phi_2, \phi_1) f^{(\mathbf{x})}(\phi_1) d\phi_1, \tag{6}$$

which is a linear transformation $L^2(\mathrm{SO}(2)) \to L^2(\mathrm{SO}(2))$. $K(\theta, r)(\phi_2, \phi_1)$ denotes the $(\phi_2, \phi_1)$ entry of the matrix at point $\mathbf{x} = (\theta, r)$, see the illustration in Figure 3-Right. We compute Equation 3 for $\rho_{\mathrm{reg}} \to \rho_{\mathrm{reg}}$ as $K(\mathrm{Rot}_\theta(\mathbf{x}))(\phi_2, \phi_1) = K(\mathbf{x})(\phi_2 - \theta, \phi_1 - \theta)$. We can use the same weight sharing scheme as in Section 4.2.

### 4.5 ANALYSIS: EQUIVARIANCE ERROR

The practical value of equivariant neural networks has been demonstrated in a variety of domains. However, theoretical analysis (Kondor & Trivedi, 2018; Cohen et al., 2019a; Maron et al., 2020) of continuous Lie group symmetries is usually performed assuming continuous functions and using the integral representation of the convolution operator. In practice, discretization can cause the model $f$ to be not exactly equivariant, with some equivariance error (EE)

$$\mathrm{EE} = \|f(T(x)) - T'(f(x))\|$$

with respect to group transformations $T$ and $T'$ of input and output respectively (Wang et al., 2020, A6). Rectangular grids are well-suited to translations, but poorly-suited to rotations. The resulting equivariance error can be so large to practically undermine the advantages of a theoretically equivariant network.

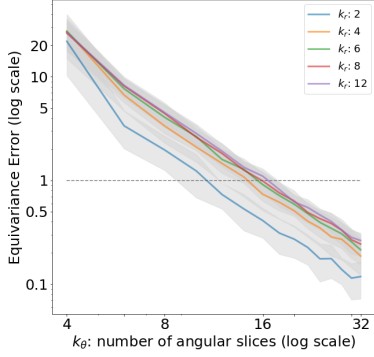

Figure 4: Experimentally, we find $k_\theta$ and expected equivariance error are inversely proportional.

Our polar-coordinate indexed circular filters are designed specifically to adapt well to the rotational symmetry. In Figure 4, we demonstrate experimentally that expected EE is inversely proportional to the number of angular slices $k_\theta$. For example, choosing $k_\theta \geq 16$ gives very low EE and does not increase the number of parameters. We also prove for $\rho_1$ features that the equivariance error is low in expectation. See Appendix A.6 for the precise statement and proof.

**Proposition.** *Let $\alpha = 2\pi/k_\theta$, and $\bar\theta$ be $\theta$ rounded to nearest value in $\mathbb{Z}\alpha$, and $\hat\theta = |\theta - \bar\theta|$. Let $F = CtsConv_{K,R}$ and $T = \rho_1(\mathrm{Rot}_\theta)$. For some constant $C$, the expected EE is bounded*

$$\mathbb{E}_{K,\mathbf{f},\mathbf{x}}[T(F(\mathbf{f}, \mathbf{x})) - F(T(\mathbf{f}), T(\mathbf{x}))] \leq |\sin(\hat\theta)|C \leq 2\pi C/k_\theta.$$

## 5 EXPERIMENTS

In this section, we present experiments in two different domains, traffic and pedestrian trajectory prediction, where interactions among agents are frequent and influential. We first introduce the statistics of the datasets and the evaluation metrics. Secondly, we compare different feature encoders and hidden feature representation types. Lastly, we compare our model with baselines.

### 5.1 EXPERIMENTAL SET UP

**Dataset** We discuss the performances of our models on (1) Argoverse autonomous vehicle motion forecasting (Chang et al., 2019), a recently released vehicle trajectory prediction benchmark, and (2) TrajNet++ pedestrian trajectory forecasting challenge (Kothari et al., 2020). For Argoverse, the task is to predict three-second trajectories based on all vehicles history in the past 2 seconds. We split 32K samples from the validation set as our test set.

**Baselines** We compare against several state-of-the-art baselines used in Argoverse and TrajNet++. We use three original baselines from (Chang et al., 2019): Constant velocity, Nearest Neighbour, and Long Short Term Memory (LSTM). We also compare with a non-equivariant continuous convolutional model, CtsConv (Ummenhofer et al., 2019) and a hierarchical GNN model VectorNet (Gao et al., 2020). Note that VectorNet only predicts a single agent at a time, which is not directly comparable with ours. We include VectorNet as a reference nevertheless.

**Evaluation Metrics**   We use domain standard metrics to evaluate the trajectory prediction performance, including (1) Average Displacement Error (ADE): the average L2 displacement error for the whole 30 timestamps between prediction and ground truth, (2) Displacement Error at $t$ seconds (DE@ts): the L2 displacement error at a given timestep $t$. DE@ts for the last timestamp is also called Final Displacement Error (FDE). For Argoverse, we report ADE and DE@ts for $t \in \{1, 2, 3\}$. For TrajNet++, we report ADE and FDE.

## 5.2   PREDICTION PERFORMANCE COMPARISON

We evaluate the performance of different models from multiple aspects: forecasting accuracy, parameter efficiency and the physical consistency of the predictions. The goal is to provide a comprehensive view of various characteristics of our model to guide practical deployment. See Appendix A.9 for an additional ablative study.

**Forecasting Accuracy**   We compare the trajectory prediction accuracy across different models on Argoverse and TrajNet++. Table 2 displays the prediction ADE and FDE comparision. We can see that ECCO with the regular representation $\rho_{\mathrm{reg}}$ achieves on par or better forecasting accuracy on both datasets. Comparing ECCO and a non-equivariant counterpart of our model CtsConv, we observe a significant 14.8% improvement in forecasting accuracy. Compare with data augmentation, we also observe a 9% improvement over the non-equivariant CtsConv trained on random-rotation-augmented dataset. These results demonstrate the benefits of incorporating equivariance principles into deep learning models.

| Model | Argoverse | | | | TrajNet++ | | #Param |
| --- | --- | --- | --- | --- | --- | --- | --- |
| | ADE | DE@1s | DE@2s | DE@3s | ADE | FDE | |
| Constant Velocity | 3.86 | 2.43 | 5.10 | 7.91 | 1.39 | 2.86 | - |
| Nearest Neighbor | 3.49 | 2.02 | 4.98 | 7.84 | 1.38 | 2.79 | - |
| LSTM | 2.13 | 1.16 | 2.81 | 4.83 | 1.11 | 2.03 | 50.6K |
| CtsConv | 1.85 | 0.99 | 2.42 | 4.32 | 0.86 | 1.79 | 1078.1K |
| CtsConv (Aug.) | 1.77 | 0.96 | 2.31 | 4.05 | - | - | 1078.1K |
| $\rho_1$-ECCO | 1.70 | 0.93 | 2.22 | 3.89 | 0.88 | 1.83 | 51.4K |
| $\rho_{\mathrm{reg}}$-ECCO | **1.62** | **0.89** | **2.12** | **3.68** | **0.84** | **1.76** | 129.8K |
| VectorNet | 1.66 | 0.92 | 2.06 | 3.67 | - | - | 72K + Decoder |

Table 2: Parameter efficiency and accuracy comparison. Number of parameters for each model and their detailed forecasting accuracy at DE@ts. CtsConv(Aug.) is CtsConv trained with rotation augmented data.

**Parameter Efficiency**   Another important feature in deploying deep learning models to embedded systems such as autonomous vehicles is parameter efficiency. We report the number of parameters in each of the models in Table 2. Compare with LSTM, our forecasting performance is significantly better. CtsConv and VectorNet have competitive forecasting performance, but uses much more parameters than ECCO. By encoding equivariance into CtsConv, we drastically reduce the number of the parameters needed in our model. For VectorNet, Gao et al. (2020) only provided the number of parameters for their encoder; a fair decoder size can be estimated based on MLP using 59 polygraphs with each 64 dimensions as input, predicting 30 timestamps, that is 113K.

**Runtime and Memory Efficiency**   We compare the runtime and memory usage with VectorNet Gao et al. (2020). Since VectorNet is not open-sourced, we compare with a version of VectorNet that we implement. Firstly, we compare floating point operations (FLOPs). VectorNet reported $n \times 0.041$ GFLOPs for the encoder part of their model alone, where $n$ is the number of predicted vehicles. We tested ECCO on a scene with 30 vehicles and approximately 180 lane marker nodes, which is similar to the test conditions used to compute FLOPs in Gao et al. (2020). Our full model used 1.03 GFLOPs versus 1.23 GFLOPs for VectorNet's encoder. For runtimes on the same test machine, ECCO runs 684ms versus 1103ms for VectorNet. Another disadvantage of VectorNet is needing to reprocess the scene for each agent, whereas ECCO predicts all agents simultaneously. For memory usage in the same test ECCO uses 296 MB and VectorNet uses 171 MB.

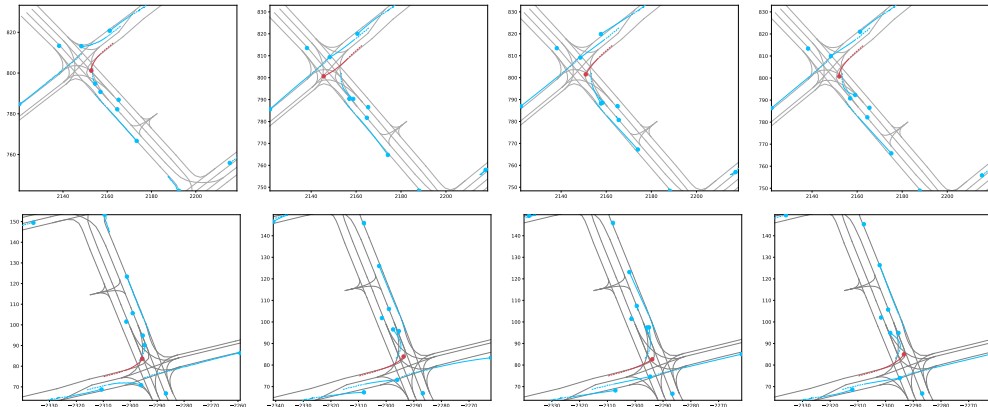

Figure 6: The x,y-axes are the position (m). The dashed line represents the 2s past trajectory. The solid line represents the 3s prediction. Red represents the agent. Top row: The predictions are made on the original data. Bottom row: We rotate the whole scene by $160°$ and make predictions on rotated data. From left to right are visualizations of ground truth, CtsConv, $\rho_1$-ECCO, $\rho_{reg}$-ECCO.

**Sample Efficiency** A major benefit of incorporating the inductive bias of equivariance is to improve the sample efficiency of learning. For each sample which an equivariant model is trained on, it learns as if it were trained on all transformations of that sample by the symmetry group (Wang et al., 2020, Prop 3). Thus ECCO requires far fewer samples to learn from. In Figure 5, we plot a comparison of validation FDE over number of training samples and show the equivariant models converge faster.

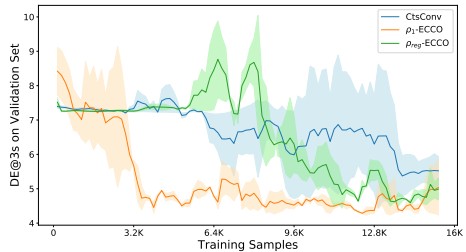

Figure 5: The learning curves on the validation set. Equivariant models converge faster using fewer samples than the non-equivariant models.

**Physical Consistency** We also visualize the predictions from ECCO and non-equivariant CtsConv, as shown in Figure 6. Top row visualizes the predictions on the original data. In the bottom row, we rotate the whole scene by $160°$ and make predictions on rotated data. This mimics the covariate shift in the real world. Note that CtsConv predicts inconsistently: a right turn in the top row but a left turn after the scene has been rotated. We see similar results for TrajNet++ (see Figure 8 in Appendix A.10).

# 6 CONCLUSION

We propose Equivariant Continuous Convolution (ECCO), a novel model for trajectory prediction by imposing symmetries as inductive biases. On two real-world vehicle and pedestrians trajectory datasets, ECCO attains competitive accuracy with significantly fewer parameters. It is also more sample efficient; generalizing automatically from few data points in any orientation. Lastly, equivariance gives ECCO improved generalization performance. Our method provides a fresh perspective towards increasing trust in deep learning models through guaranteed properties. Future directions include applying equivariance to probabilistic predictions with many possible trajectories, or developing a faster version of ECCO which does not require autoregressive computation. Moreover, our methods may be generalized from 2-dimensional space to $\mathbb{R}^n$. The orbit-stabilizer weight sharing scheme and discretized regular representation may be generalized by replacing $SO(2)$ with $SO(n)$, and polar coordinate kernels may be generalized using spherical coordinates.

## ACKNOWLEDGEMENT

This work was supported in part by Google Faculty Research Award, NSF Grant #2037745, and the U. S. Army Research Office under Grant W911NF-20-1-0334. Walters is supported by a Postdoctoral Fellowship from the Institute for Experiential AI at the Roux Institute.

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

# A  APPENDIX

## A.1  CONTINUOUS CONVOLUTION INVOLVING $\rho_{\text{reg}}$

This section is a more detailed version of Section 4.4.

Define the input $\mathbf{f}$ to be $\rho_{\text{reg}}$-field, that is, a distribution over $\mathbb{R}^2$ valued in $\rho_{\text{reg}}$. Define $K \colon \mathbb{R}^2 \to \rho_{\text{reg}} \otimes \rho_{\text{reg}}$. After identifying $SO(2)$ with its underlying manifold $S^1$, we can identify $K(\mathbf{x})$ as a map $S^1 \times S^1 \to \mathbb{R}$ and $f^{(\mathbf{x})} \colon S^1 \to \mathbb{R}$. Define the integral transform

$$K(\mathbf{x}) \odot f^{(\mathbf{x})}(\phi_2) = \int_{\phi_1 \in S^1} K(\mathbf{x})(\phi_2, \phi_1) f^{(\mathbf{x})}(\phi_1) d\phi_1.$$

For $\mathbf{y} \in \mathbb{R}^2$, define the convolution $\mathbf{g} = K \star \mathbf{f}$ by

$$\mathbf{g}(y) = \int_{x \in \mathbb{R}^2} K(x) \odot \mathbf{f}(x + y) dx.$$

The $\odot$-operation parameterizes linear maps $\rho_{\text{reg}} \to \rho_{\text{reg}}$ and is thus analogous to matrix multiplication. If we chose to restrict our choice of $\kappa$ to $\kappa(\phi_2, \phi_1) = \tilde{\kappa}(\phi_2 - \phi_1)$ for some function $\tilde{\kappa} \colon S^1 \to \mathbb{R}$ then this becomes the circular convolution operation.

The $SO(2)$-action on $\rho_{\text{reg}}$ by $\text{Rot}_\theta(f)(\phi) = f(\phi - \theta)$ induces an action on $\kappa \colon S^1 \times S^1 \to \mathbb{R}$ by

$$\text{Rot}_\theta(\kappa)(\phi_2, \phi_1) = \kappa(\phi_2 - \theta, \phi_1 - \theta).$$

This, in turn, gives an action on the torus-field $K$ by

$$\text{Rot}_\theta(K)(x)(\phi_2, \phi_1) = K(\text{Rot}_{-\theta}(x))(\phi_2 - \theta, \phi_1 - \theta).$$

Thus Equation 3, the convolutional kernel constraint, implies that $K$ is equivariant if and only if

$$K(\text{Rot}_\theta(x))(\phi_2, \phi_1) = K(x)(\phi_2 - \theta, \phi_1 - \theta).$$

We use this to define a weight sharing scheme as described in Section 3.2. The cases of continuous convolution $\rho_1 \to \rho_{\text{reg}}$ and $\rho_{\text{reg}} \to \rho_1$ may be derived similarly.

## A.2  COMPLEXITY OF CONVOLUTION WITH TORUS KERNEL

The complexity class of the convolution with torus kernel is $O(n \cdot k_{reg}^2 \cdot c_{out} \cdot c_{in})$, where $n$ is the number of particles, the regular representation is discretized into $k_{reg}$ pieces, and the input and output contain $c_{in}$ and $c_{out}$ copies of the regular representation respectively. We are not counting the complexity of the interpolation operation for looking up $K(\theta, r)$.

## A.3  EQUIVARIANT PER-PARTICLE LINEAR LAYERS

Since this operation is pointwise, unlike positive radius continuous convolution, we cannot map between different irreducible representations of $SO(2)$. Consider as input a $\rho_{\text{in}}$-field $I$ and output a $\rho_{\text{out}}$-field $O$ where $\rho_{\text{in}}$ and $\rho_{\text{out}}$ are finite-dimensional representations of $SO(2)$. We define $O^{(i)} = W I^{(i)}$ using the same $W$, an equivariant linear map, for each particle $1 \leq i \leq N$. Denote the decomposition of $\rho_{\text{in}}$ and $\rho_{\text{out}}$ into irreducible representations of $SO(2)$ as $\rho_{\text{in}} \cong \rho_1^{i_1} \oplus \ldots \oplus \rho_n^{i_n}$ and $\rho_{\text{out}} \cong \rho_1^{j_1} \oplus \ldots \oplus \rho_n^{j_n}$ respectively. By Schur's lemma, the equivariant linear map $W \colon \rho_{\text{in}} \to \rho_{\text{out}}$ is defined by a block diagonal matrix with blocks $\{W_k\}_{k=1}^n$ where $W_k$ is an $i_k \times j_k$ matrix. That is, maps between different irreducible representations are zero and each map $\rho_k \to \rho_k$ is given by a single scalar.

**Per-particle linear mapping $\rho_1 \to \rho_{\text{reg}}$ and $\rho_1 \to \rho_{\text{reg}}$.**  Since the input and output features are $\rho_1$-fields, but the hidden features may be represented by $\rho_{\text{reg}}$, we need mappings between $\rho_1$ and $\rho_{\text{reg}}$. In all cases we pair continuous convolutions with dense per-particle mappings, this we must describe per-particle mappings between $\rho_1$ and $\rho_{\text{reg}}$.

By the Peter-Weyl theorem, $L^2(\mathrm{SO}(2)) \cong \bigoplus_{i=0}^{\infty} \rho_i$. In the case of $\mathrm{SO}(2)$, this decomposition is also called the Fourier decomposition or decomposition into circular harmonics. Most importantly, there is one copy of $\rho_1$ inside of $L^2(\mathrm{SO}(2))$. Hence, up to scalar, there is a unique linear map $i_1 \colon \rho_1 \to L^2(\mathrm{SO}(2))$ given by $(a, b) \mapsto a\cos(\theta) + b\sin(\theta)$.

The reverse mapping $\mathrm{pr}_1 \colon L^2(\mathrm{SO}(2)) \to \rho_1$ is projection onto the $\rho_1$ summand and is given by the Fourier transform $\mathrm{pr}_i(f) = (\int_{S^1} f(\theta)\cos(\theta)d\theta, \int_{S^1} f(\theta)\sin(\theta)d\theta)$.

**Per-particle linear mapping** $\rho_{\mathrm{reg}} \to \rho_{\mathrm{reg}}$. Though $\rho_{\mathrm{reg}}$ is not finite-dimensional, the fact that it decomposes into a direct sum of irreducible representations means that we may take $\rho_{\mathrm{in}} = \rho_{\mathrm{out}} = \rho_{\mathrm{reg}}$ above. Practically, however, it is easier to realize the linear equivariant map $\rho_{\mathrm{reg}}^i \to \rho_{\mathrm{reg}}^j$ as a convolution over $S^1$,

$$O(\theta) = \int_{\phi \in S^1} \kappa(\theta - \phi)I(\phi)$$

where $\kappa(\theta)$ is an $i \times j$ matrix of trainable weights, independent for each $\theta$.

## A.4 Encoding Individual Particle Past Behavior

We can encode these individual attributes using a per vehicle LSTM (Hochreiter & Schmidhuber, 1997). Let $X_t^{(i)}$ denote the position of car $i$ at time $t$. Denote a fully connected LSTM cell by $h_t, c_t = \mathtt{LSTM}(X_t^{(i)}, h_{t-1}, c_{t-1})$. Define $h_0 = c_0 = 0$. We then use the concatenation of the hidden states $[h_{t_{\mathrm{in}}}^{(1)} \ \ldots \ h_{t_{\mathrm{in}}}^{(n)}]$ of all particles as $Z \in \mathbb{R}^N \otimes \mathbb{R}^k$ as the encoded per-vehicle latent features.

## A.5 Encoding Past Interactions

In addition, we also encode past interactions of particles by introducing a continuous convolution LSTM. Similar to convLSTM we replace the fully connected layers of the original LSTM above with another operation Xingjian et al. (2015). While convLSTM is well-suited for capturing spatially local interactions over time, it requires gridded information. Since the particle system we consider are distributed in continuous space, we replace the standard convolution with rotation-equivariant continuous convolutions.

We can now define $H_t, C_t = \mathtt{CtsConvLSTM}(X_t, H_{t-1}, C_{t-1})$ which is an LSTM cell using equivariant continuous convolutions throughout. Note that in this case $X_t, H_{t-1}, C_{t-1}$ are all particle feature fields, that is, functions $\{1, \ldots, n\} \to \mathbb{R}^k$.

Define CtsConvLSTM by

$$i_t = \sigma(W_{ix} \star_{cts} X_t^{(i)} + W_{ih} \star_{cts} h_{t-1} + W_{ic} \circ c_{t-1} + b_i)$$
$$f_t = \sigma(W_{fx} \star_{cts} X_t^{(i)} + W_{fh} \star_{cts} h_{t-1} + W_{fc} \circ c_{t-1} + b_i)$$
$$c_t = f_t \circ c_{t-1} + i_t \circ \tanh(W_{cx} \star_{cts} X_t^{(i)} + W_{ch} \star_{cts} h_{t-1} + b_c)$$
$$o_t = \sigma(W_{ox} \star_{cts} X_t^{(i)} + W_{oh} \star_{cts} h_{t-1} + W_{oc} \circ c_t + b_o)$$
$$h_t = o_t \circ \tanh(c_t),$$

where $\star_{cts}$ denotes CtsConv. We then can use $H_{t_{\mathrm{in}}}$ as input feature for the prediction network.

## A.6 Equivariance Error

We prove the proposition in Section 4.5.

**Proposition.** *Let $\alpha = 2\pi/k_\theta$. Let $\bar{\theta}$ be $\theta$ rounded to nearest value in $\mathbb{Z}\alpha$. Set $\hat{\theta} = |\theta - \bar{\theta}|$. Assume $n$ particles samples uniformly in a ball of radius $R$ with features $\mathbf{f} \in \rho_1^c$. Let $\mathbf{f}$ and $K$ have entries sampled uniformly in $[-a, a]$. Let the bullseye have radius $0 < R_e < R$. Let $F = CtsConv_{K,R}$ and $T_\theta = \rho_1(\mathrm{Rot}_\theta)$. Then the expected EE is bounded*

$$\mathbb{E}_{K, \mathbf{f}, \mathbf{x}}[T(F(\mathbf{f}, \mathbf{x})) - F(T(\mathbf{f}), T(\mathbf{x}))] \le |\sin(\hat{\theta})|C \le 2\pi C/k_\theta$$

*where $C = 4cna^2(1 - R_e^2/R^2)$.*

*Proof.* We may compute for a single particle $\mathbf{x} = (\psi, r)$ and multiply our result by $n$ by linearity. We separate two cases: $\mathbf{x}$ in bullseye with probability $R_e^2/R^2$ and $\mathbf{x}$ in angular slice with probability $1 - R_e^2/R^2$. If $\mathbf{x}$ is in the bullseye, then there is no equivariance error since $K(\mathbf{x})$ is a scalar matrix. Assume $\mathbf{x}$ is an angular sector.

For nearest interpolation, the equivariance error is then

$$\|\rho_1(\bar{\theta})K(\mathbf{x})\rho_1(-\bar{\theta})\rho_1(\theta)\mathbf{f} - \rho_1(\theta)K(\mathbf{x})\mathbf{f}\|.$$

Since $\rho_1(\theta)$ is length preserving, this is

$$\|\rho_1(-\theta)\rho_1(\bar{\theta})K(\mathbf{x})\rho_1(-\bar{\theta})\rho_1(\theta)\mathbf{f} - K(\mathbf{x})\mathbf{f}\|$$
$$= \|\rho_1(\beta)K(\mathbf{x})\rho_1(-\beta)\mathbf{f} - K(\mathbf{x})\mathbf{f}\| \tag{7}$$

where $\beta = \pm\hat{\theta}$. We consider only a single factor of $\rho_1$ in $\mathbf{f}$. The result will then be multiplied by $c$. Let

$$K(\mathbf{x}) = \begin{pmatrix} k_{11} & k_{12} \\ k_{21} & k_{22} \end{pmatrix}, \qquad \mathbf{f} = \begin{pmatrix} f_1 \\ f_2 \end{pmatrix}.$$

We can factor out an $a$ from $K(\mathbf{x})$ and an $a$ from $\mathbf{f}$ and assume $k_{ij}, f_i$ samples from $\text{Uniform}([-1, 1])$. One may then directly compute that Equation 7 equals

$$\sqrt{((k_{21} + k_{12})^2 + (k_{11} - k_{22})^2)(f_1^2 + f_2^2)\sin^2(\beta)}$$

This is bounded above by $4|\sin(\beta)| = 4|\sin(\hat{\theta})|$. Collecting the above factors, this proves the bound $C|\sin(\beta)|$.

The further bound follows by the first order bound,

$$|\sin(\hat{\theta})| \leq |\hat{\theta}| \leq 2\pi/k_\theta.$$

$\square$

The relationship $\text{EE} \approx 2\pi C/k_\theta$ is visible in Figure 4. We can also see clearly the significance of the term $|\sin(\hat{\theta})|$ by plotting equivariance error against $\theta$ as in Figure 7.

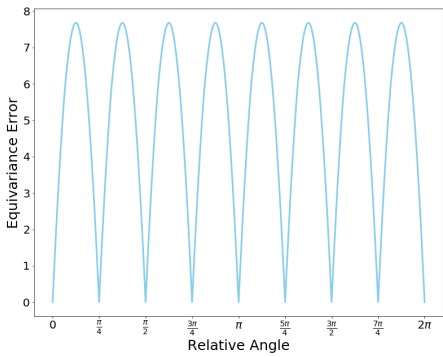

Figure 7: The above plot is generated from random input and kernels. We can clearly see the dependence of of EE on $|\sin(\hat{\theta})|$

## A.7 DATA DETAILS

Argoverse dataset includes 324K samples, which are split into 206K training data, 39K validation and 78K test set. All the samples are real data extracted from Miami and Seattle, and the dataset provides HD maps of lanes in each city. Every sample contains data for 5 seconds long, and is sampled in 10Hz frequency.

TrajNet++ Real dataset contains 200K samples. All the tracking in this dataset is captured in both indoor and outdoor locations, for example, university, hotel, Zara, and train stations. Every sample in this dataset contains 21 timestamps, and the goal is to predict the 2D spatial positions for each pedestrain in the future 12 timestamps.

## A.8 IMPLEMENTATION DETAILS

Argoverse dataset is not fully observed, so we only use cars with complete observation as our input. Since every sample doesn't include the same number of cars, we only choose those scenes with less than or equal to 60 cars and insert dummy cars into them to achieve consistent car numbers. TrajNet++ Real dataset is also not fully observed. And here we keep our pedestrain number consistent to 160.

Moreover, for each car, we use the average velocity in the past 0.1 second as an approximate to the current instant velocity, i.e. $v_t = (p_t - p_{t-1})/2$. As for map information, we only include center lanes with lane directions as features. Also, we introduce dummy lane node into each scene to make lane numbers consistently equal to 650.

In TrajNet++ task, no map information is included. And since pedestrians don't have a speedometers to tell them exactly how fast they are moving as drivers, instead they depends more on the relative velocities and relative positions to other pedestrians, we tried different combination of features in ablative study besides only using history velocities.

Our models are all trained by Adam optimizer with base learning rate 0.001, and the gamma rate for linear rate scheduler is set to be 0.95. All our models without map information are trained for 15K iterations with batch size 16 and learning rate is updated every 300 iterations; for models with map information, we train them for 30K iterations with batch size 16 and learning rate is updated every 600 iterations.

For CtsConv, we set the layer sizes to be 32, 64, 64, 64, and kernel size $4 \times 4 \times 4$; for $\rho_1$-ECCO, the layer sizes are 16, 32, 32, 32, $k_\theta$ is 16, $k_r$ is 3; for $\rho_{\mathrm{reg}}$-ECCO, we choose layer size 8, 16, 8, 8, $k_\theta$ 16, $k_r$ 3, and regular feature dimension is set to be 8. For Argoverse task, we set the CtsConv radius to be 40, and for TrajNet++ task we set it to be 6.

## A.9 ABLATIVE STUDY

We perform ablative study for ECCO to further diagnose different encoders, usage of HD maps and other model design choices.

**Choice of encoders**   Unlike fluid simulations (Ummenhofer et al., 2019) where the dynamics are Markovian, human behavior exhibit long-term dependency. We experiment with three different encoders refered to as Enc to model such long-term dependency: (1) concatenating the velocities from the past $m$ frames as input feature, (2) passing the past velocities of each particle to the same LSTM to encode individual behavior of each particle, and (3) implementing continuous convolution LSTM to encode past particle interactions. Our continuous convolution LSTM is similar to convLSTM (Xingjian et al., 2015) but uses continuous convolutions instead of discrete gridded convolutions.

We use different encoders to time-aggregate features and compare their performances (Table 3).

**Use of HD Maps**   In Table 4, we compare performance with and without map input features.

**Choice of features for pedestrian**   Unlike vehicles, people do not have a velocity meter to tell him how fast they actually walk. We realize that people actually tend to adjust their velocities based on others' relative velocity and relative position. We experiment different combination of features (Table 5), finding using relative velocities and relative positions as feature has the best performance.

## A.10 QUALITATIVE RESULTS FOR TRAJNET++

Figure 8 show qualitative results for TrajNet++. Note that the non-equivariant baseline (2nd column) depends highly on the global orientation whereas the ground truth and equivariant models do not.

| Encoder | Argoverse | | | | TrajNet++ | |
|---|---|---|---|---|---|---|
| | ADE | DE@1s | DE@2s | DE@3s | ADE | FDE |
| Markovian | 4.67 | - | - | 9.84 | 0.969 | 1.952 |
| LSTM | 2.05 | 1.06 | 2.51 | 4.71 | 0.909 | 1.909 |
| CtsConvLSTM | 3.98 | 2.02 | 5.11 | 8.40 | 0.962 | 1.941 |
| CtsConvDLSTM | 2.02 | 1.03 | 2.46 | 4.58 | 0.910 | 1.916 |
| D-Concat(20t feats) | **1.87** | **1.01** | **2.43** | **4.22** | **0.895** | **1.872** |

Table 3: Ablation study on encoders for Argoverse and TrajNet++. Markovian: Use the velocity from the most recent time step as input feature. LSTM: Used LSTM to encode velocities of 20 timestamps. CtsConvLSTM: Instead of dense layer, the gate functions in LSTM are replaced by CtsConv. CtsConvDLSTM: Replaced gate functions by CtsConv + Dense. D-Concat (20t feats): Stacked velocities of 20 time steps as input.

| Model | w/o Map | | | | w/ Map | | | |
|---|---|---|---|---|---|---|---|---|
| | ADE | DE@1s | DE@2s | DE@3s | ADE | DE@1s | DE@2s | DE@3s |
| CtsConv | 1.87 | 1.01 | 2.43 | 4.22 | 1.85 | 0.99 | 2.42 | 4.32 |
| $\rho_1$-ECCO | **1.81** | 1.02 | 2.42 | 4.14 | 1.70 | 0.93 | 2.22 | 3.89 |
| $\rho_{\text{reg}}$-ECCO | **1.81** | **1.00** | **2.38** | **4.12** | **1.62** | **0.89** | **2.12** | **3.68** |

Table 4: Ablative study on HD maps for Argoverse. Prediction accuracy comparison with and without HD Maps.

| Velocity | Relative Position | Acceleration | ADE | FDE |
|---|---|---|---|---|
| Absolute | ✕ | ✕ | 0.92 | 1.95 |
| Absolute | ✕ | ✓ | 0.90 | 1.87 |
| Relative | ✕ | ✓ | 0.89 | 1.86 |
| Relative | ✓ | ✓ | **0.86** | **1.79** |

Table 5: Ablative study on features for Traj++. Acceleration means whether we used acceleration to make numerically extrapolated position.

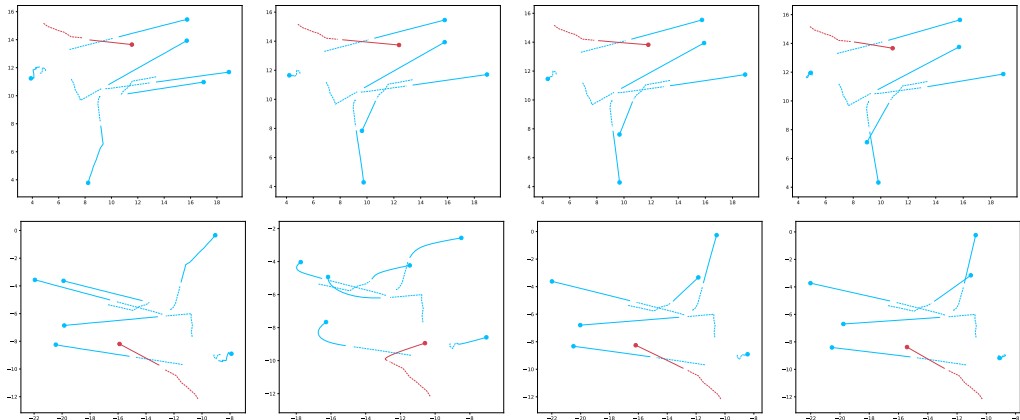

Figure 8: The x,y-axes are the position (m). The dashed line represents the 2s past trajectory. The solid line represents the 3s prediction. Red represents the agent. Top row: The predictions are made on the original data. Bottom row: We rotate the whole scene by 160° and make predictions on rotated data. From left to right are visualizations of ground truth, CtsConv, $\rho_1$-ECCO, $\rho_{\text{reg}}$-ECCO.

