# OpenReview forum: "Trajectory Prediction using Equivariant Continuous Convolution"
_ICLR.cc/2021/Conference — ICLR 2021 Poster_

### Official Review · AnonReviewer3 · 2020-10-17
**Interesting work but with flaws**

**Rating:** 6
**Confidence:** 4

**Review:**

--- Summary

This paper presents a novel Equivariant Continous COnvolution (ECCO) method for vehicle and pedestrian trajectory prediction. ECCO extends the previous continuous convolution method and makes it rotationally equivariant. To achieve that, they constrain the convolution kernel function K and make only the K(0, r) component freely trainable, and use it to derive the other components. They also propose a torus kernel that can be used on functions on circles. They evaluated their approach on two real-world trajectory prediction datasets, Argoverse and TrajNet++. They compared to basic baselines including constant velocity, nearest neighbor, and LSTM, as well as CtsConv (non-equivariant continuous convolution) and VectorNet. Results show that ECCO achieves significantly lower prediction errors than CtsConv and has fewer model weights. However, their prediction error is slightly higher than VectorNet at 2s and 3s.


--- Strengths

- The proposed ECCO trajectory prediction approach is novel in that the previous approaches are either not rotationally equivariant or only run on a single actor.

- Making convolution operations rotationally equivariant can be very useful to many applications where rotations on the input aren't supposed to affect the outputs. The ECCO method proposed in this paper is very general, and I think it can be applied to the other applications as well.

- With weight-sharing, the ECCO method greatly reduces the number of weights. The evaluation result also shows that the reduction of the trainable weights makes the model more data efficient.


--- Major issues and suggestions

- Actually, most of the state-of-the-art trajectory prediction approaches are already rotationally equivariant because they usually rotate their input scene to have the target actor that they are predicting always face up (e.g., in VectorNet). I think the only non-equivariant approaches are those one-shot approaches that predict all actors in the scene simultaneously. This work is interesting in that it is both one-shot and rotationally equivariant. However, from the evaluation results, it is not clear to me why predicting all actors in the scene simultaneously is a desired property. ECCO in fact has higher prediction errors than VectorNet. The inference speed can be a potential win, but the inference speed numbers are shown in the evaluation, unfortunately.

- The evaluation baseline setup paragraph says "VectorNet only predicts a single agent at a time, which is not directly comparable with ours". I don't get why it is not directly comparable. In fact, VectorNet has lower prediction errors than ECCO at 2s and 3s. If the inference speed is an issue, the paper should show it with numbers.

- The "Parameter Efficiency" paragraph says "The number of parameters has strong implications in memory usage, battery consumption and compute speed." Why not just measure and show the memory usage and compute speed numbers?

- The paper claims that trajectories predicted by the ECCO method are more physically realistic, but I don't see any results in the paper demonstrating this.


--- Other comments

- Section 3.2 is a little hard to understand. For example, what is S^1, what is GL, and what is L^2(SO(2))? I think providing more background or some references would be useful.


--- Response to author's answers

> The “target actor points up” method can be used to predict multiple agents by running repeatedly, these predictions are each independent, whereas in reality future trajectories for vehicles are mutually dependent. Two cars may have independently likely trajectories, but if they intersect, they are unlikely to occur together.

Yes, this is a commonly used argument used by works that predict all actors in the scene simultaneously. However, whether the claimed benefit is achieved by the proposed model is yet to be proved. The fact that the proposed model has higher prediction errors than the state-of-the-art single-actor prediction model makes me doubt whether this is true.

Thank you for updating the paper with the inference time numbers. These results are really useful.

> For runtimes on the same test machine, ECCO runs 684ms versus 1103ms for VectorNet.

1103ms is really high a latency for any real-time robotics system. For completeness, could you also include the number of actors that are predicted by VectorNet in 1103ms?

---

> ### Author Response · Authors · 2020-11-24
> **Response to Reviewer 3**
>
> Thank you for taking the time to understand and comment on our paper.  We are glad you have highlighted one of the major strengths of our work which is that it is both rotationally equivariant and runs simultaneously for all actors.  We address your specific questions below.
>
> >However, from the evaluation results, it is not clear to me why predicting all actors in the scene simultaneously is a desired property.
>
> **Computational and Sample Efficiency:**   Since our hidden features are rotatable vectors, we can reuse the same features for predicting any agent in the scene.  This is similar to multitask learning in which features can be transferred from one task to another.  This leads to better sample efficiency.  The “target actor points up” method would need to be trained on the same scene repeatedly for each actor to learn what our model can effectively learn in one pass.
>
> **Correlated Predictions:** The “target actor points up” method can be used to predict multiple agents by running repeatedly, these predictions are each independent, whereas in reality future trajectories for vehicles are mutually dependent.  Two cars may have independently likely trajectories, but if they intersect, they are unlikely to occur together.
>
> >ECCO in fact has higher prediction errors than VectorNet.  The inference speed can be a potential win, but the inference speed numbers are shown in the evaluation, unfortunately.
>
>  VectorNet is fairly specialized to predicting road traffic.  Their methods use the fact road information may be summarized using polylines which track the lanes.  **Our method is more general as shown by our TrajNet++ (pedestrian trajectory) experiments.**
>
> Inference time comparison is a good idea.  VectorNet is not open source, so we 1) compare FLOPs versus those reported in the VectorNet paper, 2) compare runtimes using a version of the VectorNet code we implement.  Vectornet reports $n \times 0.041$ GFLOPs for their encoder where $n$ is the number of predicted vehicles.  They do not include the FLOPs of the decoder.  We tested our model on scene with 30 vehicles and approximately 180 lane marker nodes, which we judge to be similar to the test conditions used to compute FLOPs in the VectorNet paper.  Our full model used 1.03 GFLOPs versus 1.23 GFLOPs for VectorNet’s encoder.  For runtimes on the same test machine, our model runs 684ms vs. 1103ms for VectorNet.  The disadvantage of VectorNet is needing to reprocess the scene for each agent, whereas ours predicts all agents simultaneously.
>
> >The evaluation baseline setup paragraph says "VectorNet only predicts a single agent at a time, which is not directly comparable with ours". I don't get why it is not directly comparable.
>
> You are correct, we can compare VectorNet and ECCO on either single-agent or multi-agent predictions (by running VectorNet repeatedly for each agent).  However, the numbers we and VectorNet report are only for the single car that is labeled “agent” in the argoverse dataset, which is the standard metric for this dataset.
>
> >Why not just measure and show the memory usage and compute speed numbers?
>
> See above for walltime and GFLOPs.  For memory usage our model uses 296 MB and VectorNet uses 171MB in the same test as above.
>
> >The paper claims that trajectories predicted by the ECCO method are more physically realistic, but I don't see any results in the paper demonstrating this.
>
> We mean the forecasting *function* is more physically realistic.  The system dynamics are invariant to the choice of coordinate frame.  Consider Figure 6: whether the non-equivariant model (column 2) predicts a left or right turn depends on coordinate rotation.  The equivariant model (column 3 or 4) respects rotational invariance.
>
> >Section 3.2 is a little hard to understand. For example, what is S^1, what is GL, and what is L^2(SO(2))? I think providing more background or some references would be useful.
>
> We have added references and some additional clarification to the paper.  Briefly, $S^1 = \lbrace (x,y) : x^2+y^2 = 1 \rbrace$ is the 1-dimensional manifold of a circle.   $GL(\mathbb{R}^n)$ is the set of $n$-by-$n$ invertible matrices.  $SO(2)$ is the abstract group of rotations of the plane.  As a set $SO(2) = \lbrace Rot_\theta : \theta \in [0,2\pi) \rbrace$.  The set $L^2(X)$ is the space of square integrable functions on $X$, i.e. $L^2(X) = \lbrace f \colon X \to \mathbb{R} : |f|^2 \text{ is integrable} \rbrace$.  So $L^2(SO(2))$ is the set of square integrable functions $SO(2) \to \mathbb{R}$.

---

### Official Review · AnonReviewer4 · 2020-10-28
**A Well-written paper, but the idea is not new.**

**Rating:** 6
**Confidence:** 2

**Review:**

The polar-coordinate convolution has been proposed by multiple previous literatures, e.g. “Polar Transformer Networks” and “Rotational Rectification Network: Enabling Pedestrian Detection for Mobile Vision”.

For self-driving system, one should be able to get the global pose of your vehicle, then correct the global rotation of your scene based on the global pose. This should be a more practical solution to this issue. To me, this paper is overcomplicated as of a solution to the problem.

The theory part is well-written and easy to understand.

Experiment is not strong enough though. I want to see how each of your variances add up in increased accuracy in ECCO network, e.g. experiments about stacking up “Weight sharing by orbits and stabilizers”, “polar coordinate kernels” and “hidden layers as regular representations”. Also, for the final model comparison (Table 2), I would like to see how ECCO compares to state-of-the-art other than VectorNet.

---

> ### Author Response · Authors · 2020-11-24
> **Response to Reviewer 4: misunderstanding and clarification of novelty**
>
> Thank you for reviewing our paper and for constructive comments.  We are glad you found our theory well-written and understandable.
>
> > The polar-coordinate convolution has been proposed by multiple previous literatures, e.g. “Polar Transformer Networks” and “Rotational Rectification Network: Enabling Pedestrian Detection for Mobile Vision”.
>
> Thank you for these references which we happily add as related works, however, there are **major differences with our approach**.  While our method and PTN both make use of polar coordinates and evenly sample points on the circle, significant differences include: 1) our method operators on point clouds which are more general than 2D grids of PTN; 2) our model is more expressive since we assign feature vectors at each point in our point cloud, whereas in PTN, feature vectors are computed at only the origin; 3) our model is also more flexible since our feature vectors can live in any SO(2)-representation, not only the regular representation.   Rotational Rectification Networks is even further from our work in that they achieve equivariance through a single rotational canonicalization whereas each linear layer of our network is inherently equivariant.  Moreover our filters account for local symmetry in the problem as each car may have its own preferred frame of reference.
>
> >For self-driving system, one should be able to get the global pose of your vehicle, then correct the global rotation of your scene based on the global pose. This should be a more practical solution to this issue. To me, this paper is overcomplicated as of a solution to the problem.
>
> Yes, for a single vehicle, we could use this approach (which is used by VectorNet).  However, it cannot predict multiple agents at once. **Our method is both rotationally equivariant and multiagent**.  Though “global pose” can be used to predict multiple agents by running repeatedly and orienting to each new agent, these predictions are each independent, whereas in reality the probability of future trajectories for vehicles is strongly mutually dependent.
>
> An alternative solution would be to use “local pose” and rotate for every vehicle for each convolution.  However, this is inefficient and our method of rotationally equivariant filters automatically adapts to “local pose” without any rotation.
>
> Lastly, our method has better sample efficiency.  The “global pose” method would need to be trained on the same scene repeatedly for each actor to learn what our model can effectively learn in one pass.  Since our hidden features are rotatable vectors, we can reuse the same features for predicting any agent in the scene.  This is similar to multitask learning in which features can be transferred from one task to another.
>
> > I want to see how each of your variances add up in increased accuracy in ECCO network, e.g. experiments about stacking up “Weight sharing by orbits and stabilizers”, “polar coordinate kernels” and “hidden layers as regular representations”. Also, for the final model comparison (Table 2), I would like to see how ECCO compares to state-of-the-art other than VectorNet.
>
> We agree ablative comparison is essential, and we already compare the suggested 3 aspects in Table 2: CtsConv uses none, $\rho_1$-ECCO uses “Weight sharing by orbits and stabilizers” and “polar coordinate kernels”, and $\rho_{reg}$-ECCO uses all 3.  We do not test “polar coordinates” and “Weight sharing by orbits and stabilizers” separately since our use of polar coordinates is only to facilitate weight sharing.  We have also added an ablative study for our feature choices for TrajNet++ (Table 4, Appendix A.8) and a study comparing encoders for TrajNet++ (Table 3, Appendix A.8).
>
> We would like to run other baselines including the recently proposed LANE-GCN [D1], but we do not have access to the code since it is not yet open source.  While it is worth comparing to baselines such as VectorNet and LANE-GCN, it is worth noting both are specialized to the map structure of roads specifically, whereas our method is more general.
>
> [D1] Liang, Ming, Bin Yang, Rui Hu, Yun Chen, Renjie Liao, Song Feng, and Raquel Urtasun. "Learning lane graph representations for motion forecasting." In European Conference on Computer Vision, pp. 541-556. Springer, Cham, 2020.

---

### Official Review · AnonReviewer2 · 2020-10-29

**Rating:** 7
**Confidence:** 3

**Review:**

Summary:

The paper proposes a continuous convolution equivariant with respect to SO2 for trajectory prediction.
Rotational equivariance is achieved by a weight sharing scheme within kernels in polar coordinates.
The proposed scheme approximates rotational equivariance well and has similar or better performance on
the trajectory prediction tasks than models without equivariance while using fewer parameters.



Score:

I tend to accept for this paper. Rotational equivariance is an important problem that can hinder the broad application of ConvNets to sparse and continuous data where often graph networks are used.
Trajectory prediction is a good application for rotational equivariance and the paper is written very general allowing the method to be transferred to other problems.
Although some interesting comparisons with existing methods are missing, overall the experiments can show well the advantages of the proposed convolutions and are convincing.



Strengths:

Combining weight sharing with interpolation is a good idea to avoid unnecessary evaluations of the kernel at runtime.

Figure 3 is very helpful but the font size for the labels should be optimized.

The experiments about physical consistency and the equivariance error answer important questions about the method.



Weaknesses:

There are no direct comparisons to other equivariant convolutions, e.g. a comparison with Tensor field networks.
Although the presented method is specialized for SO2, a comparison would be helpful to demonstrate the efficiency of the implementation.

There is also no information about the runtime and memory efficiency of the convolutions.
Table 2 shows the number of parameters but the different models may have a different memory footprint during inference.

The Background section is a bit short. Adding some references to the literature on rotational equivariance in this section would be helpful.



Questions:

A7 mentions that all models are trained for the same number of iterations.
Did all models converge within 15k/30k iterations?
If not what is the performance of each model after convergence?

Are the models without equivariant convolutions trained with data augmentations?

What is the complexity class of the convolution with torus kernel?

---

> ### Author Response · Authors · 2020-11-24
> **Response to Reviewer 2: thank you and added comparisons**
>
> Thank you for your careful reading of our paper.  We are glad you found our method to be a solution to an important problem and that you see it as a general and more broadly applicable method, since this was indeed one of our goals.  We appreciate your suggestions for strengthening the paper and have acted on them where possible.
>
> >Figure 3 is very helpful but the font size for the labels should be optimized.
>
> We will improve the labels in our updated paper.
>
> > There are no direct comparisons to other equivariant convolutions, e.g. a comparison with Tensor field networks.
>
> This is a good idea.  However, since TFNs are 3D, application to 2D requires reformulation in terms of circular harmonics and $SO(2)$ Clebsch-Gordan coefficients, which is non-trivial.  Alternatively, we considered applying TFNs by embedding our data in a plane in 3D, but $SO(3)$-symmetry on a plane allows for reflections, which are not a symmetry of our system due to differences of left/right in driving. (Flipping the plane is a rotation in 3D.)
>
> >There is also no information about the runtime and memory efficiency of the convolutions.
>
> We agree these comparisons would be good to include.  VectorNet is not open source, so we 1) compare FLOPs versus those reported in the VectorNet paper, 2) compare runtimes using a version of the VectorNet code we implement.  Vectornet reports $n \times 0.041$ GFLOPs for their encoder where $n$ is the number of predicted vehicles.  They do not include the FLOPs of the decoder.  We tested our model on scene with 30 vehicles and approximately 180 lane marker nodes, which we judge to be similar to the test conditions used to compute FLOPs in the VectorNet paper.  Our full model used 1.03 GFLOPs versus 1.23 GFLOPs for VectorNet’s encoder.  For runtimes on the same test machine, our model runs 684ms vs. 1103ms for VectorNet.  The disadvantage of VectorNet is needing to reprocess the scene for each agent, whereas ours predicts all agents simultaneously.  A future work which might be even faster would be a version of our model without autoregressive computation.  For memory usage in the same test our model uses 296 MB and VectorNet uses 171 MB.  On its own, weight sharing saves parameters, but not memory.
>
> >The Background section is a bit short. Adding some references to the literature on rotational equivariance in this section would be helpful.
>
> We agree some additional description of previous methods used to enforce rotational equivariance would improve the background section, and have expanded it to include  steerable CNN (Cohen & Welling, 2016b), specifically E2CNN (Weiler & Cesa, 2019), and results of (Kondor & Trivedi) and (Cohen et al 2019). Tensor field networks (Thomas et al. 2018) and harmonic networks (Worrall et al. 2017) are contrasted in the method section.
>
> >  Did all models converge within 15k/30k iterations?
>
> Yes, all our models presented in this paper actually converged within 15k/30k, the reason why we chose this number is right to make sure all the models could converge.
>
> >Are the models without equivariant convolutions trained with data augmentations?
>
> Our continuous convolution baseline is not trained with augmentations.  Doing so would be a good comparison.  ~~We will try to perform it before the review window closes.~~ We have added this comparison.  See below or Table 2 for results.
>
> >What is the complexity class of the convolution with torus kernel?
>
> Convolution with a torus kernel is $ O( n \cdot k_{reg}^2 \cdot c_{out} \cdot c_{in}) $, where $n$ is the number of particles, the regular representation is discretized into $k_{reg}$ pieces, and the input and output contain $c_{in}$ and $c_{out}$ copies of the regular representation respectively. We are not counting the complexity of the interpolation operation for looking up $K(\theta,r)$.

---

> > ### Author Response · Authors · 2020-11-24
> > **Comparison to data augmentation**
> >
> > We have added a comparison to the non-equivariant CtsConv baseline trained with additional rotationally augmented samples.  As expected this improves performance versus no augmentation, but does not match the equivariant models.   The updated results can be found in Table 2.  For a quick summary comparison, for Argoverse, the ADEs are CtsConv: 1.85, CtsConv(Aug): 1.77, $\rho_{reg}$-ECCO: 1.62.  The other metrics show a similar pattern.

---

### Official Review · AnonReviewer1 · 2020-11-02
**Writing should be improved, many details missing**

**Rating:** 5
**Confidence:** 2

**Review:**

The authors present a novel approach to trajectory prediction, where they use rotationally-invariant continuous convolutions. They present background on such convolutions and discuss a way to efficiently store and compute them,
I found the paper not really well written, with a number of notation not being explained or skipped. I went over several relevant citations that the authors provided which helped, but some parts still remain unclear to me. I may not have understood all the math presented, which is the reason I lowered my confidence.

More comments:
- The writing could be improved significantly. The authors introduce a lot of notation and concepts without really defining them well. E.g., K goes from K(.) to K(.,.) to K(.)(.,.), and similarly for rho_1 and rho_reg. I may be missing some background knowledge here, but I do feel that this should be much smoother and better explained nevertheless.
- E.g., here they also introduce phi, theta, and other notation, but what that corresponds to in the trajectory prediction aspect is not explained or discussed. Or T in Section 4.5.
- The authors mention physically-consistent predictions, but the method doesn't really contribute directly to that aspect of the prediction problem. Nor is it evaluated and discussed deeper. This should be removed or discussed more.
- Same goes for "transparency", it is not really discussed deeper.
- The authors don't explain well what are the actual model inputs that they use. This is glossed over, while usually works spend a lot of time here. They just say that they use histories and map info, but how exactly is unclear.
- The authors also say that, unlike some competitors, they predict for all actors. This was also unclear how exactly. Or how are interactions taken into account. The explanations are very vague indeed.
- Why are the scenes in Fig 6 not exactly the same, some actor locations are different? And especially for CtsConv where the actor is completely different it seems?
- Curious how was the training done? Did you have to do some augmentation with scene rotations and imposing of the constraints?
In general, the discussion can be improved significantly. I marked my grade as 5 as I don't want to block the paper just because I didn't understand all the items, but I would be fine with mark 4 as well assuming the other reviewers have concerns along similar lines as myself.

---

> ### Author Response · Authors · 2020-11-24
> **Response to Reviewer 1: More Details Added**
>
> We are glad you found our approach novel.  Thank you for your detailed comments pointing out areas where the clarity of our paper can be improved.  We have included clarifications below and we will also put them in the paper.   We appreciate the time you took to look into related works.
>
> >The authors introduce a lot of notation and concepts without really defining them well.
>
> We have made the definitions and notations clearer.  To specifically address some of the confusing notation you point out: The distinction between $K(\mathbf{x})$ and $K(\theta,r)$ is whether we write the vector $\mathbf{x} = (\theta,r)$ in polar coordinates or not.   Note that the output $K(\mathbf{x})$ is itself a \emph{matrix} and thus we may index into this matrix as such $K(\mathbf{x})(\phi_1,\phi_2)$ as we do in the paragraph “Convolution with Torus Kernel” and in Figure 3.  The reason we use $\phi_1, \phi_2$ as indices instead of $i,j$ is because for a torus kernel both indices are continuous and ``"wrap around," and so we think of the indices as angles on a circle.  We also write the matrix as a function of its indices using parentheses instead of subscripts.  This is the logical choice from a representation theory standpoint since the group $\mathrm{SO}(2)$ acts naturally on functions on the circle.
>
> > they also introduce phi, theta, and other notation, but what that corresponds to in the trajectory prediction aspect is not explained
>
> We have tried to consistently use $\theta$ for rotation transformation and $\phi$ when indexing torus kernels.  From a driver’s perspective, one may think of $\theta$ as the angle pointing to a nearby vehicle and $\phi$ as the direction of that vehicle’s velocity.   $T$ and $T’$ are the group transformations of input and output respectively.  We have clarified this notation.
>
> >The authors mention physically-consistent predictions, but the method doesn't really contribute directly to that aspect of the prediction problem.
>
> One central tenet of physics is that the system should be independent of the choice of coordinates.  For us, this means the model should give consistent outputs for intersections rotated into a different orientation.  Figure 6 shows that our model follows this physical law (column 3/4), whereas the prediction of the baseline model (column 2) depends on the orientation of the system and is thus physically inconsistent.
>
> >Same goes for "transparency", it is not really discussed deeper.
>
> We have removed the claim of “transparency,” from the paper.  Rather equivariance improves trustworthiness of the model.   While not every aspect of the model output is guaranteed or predictable, our constraints provide a guarantee our model will follow the above physical law.
>
> >The authors don't explain well what are the actual model inputs that they use.
>
> The feature of each car is $[v_1, v_2, …, v_t]$, where each $v_i$ is the instant velocity calculated by $(p_i - p_{i-1})/2$. More detailed comparison for the choice for the car feature encoder could be found in Appendix A.7. As for the map information, Argoverse provided the key points of each centerline from the splines at the same spatial distance, and we used the lane direction vector as the feature of each key point, which is calculated by $(p_i - p_{i-1})$. We will clarify this in the paper.
>
> >The authors also say that, unlike some competitors, they predict for all actors. This was also unclear how exactly.
>
> This is a key difference between our model and e.g. VectorNet.  Our model outputs the velocity of every vehicle in the scene at the next timestep.  By concatenating this prediction to the previous input timesteps and applying the model again, we can predict the future velocities of all vehicles as far as desired into the future.  That is, we use autoregression.
>
> >Or how are interactions taken into account.
>
> Interactions are taken into account using the continuous convolution operation.  Continuous convolution may be thought of as a message-passing algorithm similar to graph neural networks.  The inputs, hidden layers, and outputs are all features centered at the vehicles, which may be thought of as nodes.  At each round of continuous convolution, we apply weights to all the features of all nearby vehicles and then aggregate them by summing.  The difference versus graph convolution is that the weights depend on the location of the nearby vehicles and the edges in the graph representing nearby vehicles change as the vehicles move.  We have highlighted this point in the paper.

---

> > ### Author Response · Authors · 2020-11-24
> > **Response to Reviewer 1, Part 2**
> >
> > >Why are the scenes in Fig 6 not exactly the same, some actor locations are different? And especially for CtsConv where the actor is completely different it seems?
> >
> > The dots mark the predicted ending position of the actors. The difference in the predicted actor’s position highlights the lack of physical consistency for the **non-equivariant** CtsConv.  In the bottom row the inputs have all been rotated by 160 degrees, but as you can see the non-equivariant model (CtsConv) does not predict consistent outputs.  Thus, not only is the model not constrained to be equivariant, it does not learn to be equivariant either.  This provides evidence for the need to encode this as a model constraint.
> >
> > >Curious how was the training done?
> >
> > Our models are all trained by Adam optimizer with base learning rate 0.001, and the gamma rate for linear rate scheduler is set to be 0.95. All our models without map information are trained for 15K iterations with batch size 16 and learning rate is updated every 300 iterations; for models with map information, we train them for 30K iterations with batch size 16 and learning rate is updated every 600 iterations.
> >
> > >Did you have to do some augmentation with scene rotations and imposing of the constraints?
> >
> > We did not do any augmentation or scene rotation. In principle, $G$-equivariant models do not require augmentations by transformations in $G$ [A1].  Our rotationally equivariant network learns from a single sample the same as though being trained on all rotated versions of that sample.
> >
> > [A1] Cohen, T. and Welling, M., 2016, June. Group equivariant convolutional networks. In International conference on machine learning (pp. 2990-2999).

---

### Author Response · Authors · 2020-11-24
**Summary and Revisions**

Thank you to all our reviewers.  In response to your questions and suggestions we have made several changes which we feel strengthen our paper.  We have uploaded a revised draft with changes marked in blue.   Our changes include:

- Added FLOPs, memory, and runtime comparisons between our model and VectorNet
- Added ablative study for TrajNet++ features
- Added encoder comparison study for TrajNet++
- Added comparison to data-augmented baseline
- Clarified notation throughout paper
- Extended the background section with explanation and references on rotational equivariance
- Added Polar Transformer Networks and Rotational Rectification Networks to related works

---

### Decision · Program_Chairs · 2021-01-07
**Final Decision**

**Decision:**

Accept (Poster)

**Comment:**

The paper proposes and studies a new SO(2)-equivariant convolution layer for vehicle and pedestrian trajectory prediction. The experiments are detailed and demonstrate the effectiveness of the approach in relation to non-equivariant models.